# Doubly Robust Bias Reduction in Infinite Horizon Off-Policy Estimation

**Ziyang Tang** [*]
The University of Texas at Austin
ztang@cs.utexas.edu

**Yihao Feng** [*]
The University of Texas at Austin
yihao@cs.utexas.edu

**Lihong Li**
Google Research
lihong@google.com

**Dengyong Zhou**
Google Research
dennyzhou@google.com

**Qiang Liu**
The University of Texas at Austin
lqiang@cs.utexas.edu

## Abstract

*Infinite horizon* off-policy policy evaluation is a highly challenging task due to the excessively large variance of typical importance sampling (IS) estimators. Recently, Liu et al. (2018a) proposed an approach that substantially reduces the variance of infinite horizon off-policy evaluation by estimating the stationary density ratio, but at the cost of introducing biases due to errors in density ratio estimation. In this paper, we develop a bias-reduced augmentation of their method, which can take advantage of a learned value function to improve accuracy. Our method is *doubly robust* in that the bias vanishes when either the density ratio or value function estimation is perfect. In general, when either of them is accurate, the bias can also be reduced. Both theoretical and empirical results show that our method yields significant advantages over previous methods.

## 1 Introduction

A key problem in reinforcement learning (RL) (Sutton & Barto, 1998) is off-policy policy evaluation: given a fixed *target policy* of interest, estimating the average reward garnered by an agent that follows the policy, by only using data collected from different *behavior policies*. This problem is widely encountered in many real-life applications (e.g., Murphy et al., 2001; Li et al., 2011; Bottou et al., 2013; Thomas et al., 2017), where online experiments are expensive and high-quality simulators are difficult to build. It also serves as a key algorithmic component of off-policy policy optimization (e.g., Dudík et al., 2011; Jiang & Li, 2016; Thomas & Brunskill, 2016; Liu et al., 2019b).

There are two major families of approaches to policy evaluation. The first approach is to build a simulator that mimics the reward and next-state transitions of the real environment (e.g., Fonteneau et al., 2013). While straightforward, this approach strongly relies on the model assumptions in building the simulator, which may invalidate evaluation results. The second approach is to use importance sampling to correct the sampling bias in off-policy data, so that an (almost) unbiased estimator can be obtained (Liu, 2001; Strehl et al., 2010; Bottou et al., 2013). A major limitation, however, is that importance sampling can become inaccurate due to high variance. In particular, most existing IS-based estimators compute the weight as the product of the importance ratios of many steps in the trajectory, causing excessively high variance for problems with long or infinite horizon, yielding a *curse of horizon* (Liu et al., 2018a).

Recently, Liu et al. (2018a) proposes a new estimator for infinite-horizon off-policy evaluation, which presents significant advantages to standard importance sampling methods. Their method directly estimates the density ratio between the state stationary distributions of the target and behavior policies, instead of the trajectories, thus avoiding exponential blowup of variance in the horizon. While Liu et al.'s method shows much promise by significantly *reducing the variance*, in practice, it may suffer from *high bias* due to the error or model misspecficiation when estimating the density ratio function.

---

[*]The first two authors contributed equally to this work.

In this paper, we develop a *doubly robust* estimator for infinite horizon off-policy estimation, by integrating Liu et al.'s method with information from an additional value function estimation. This significantly reduces the bias of Liu et al.'s method once either the density ratio, or the value function estimation is accurate (hence doubly robust). Since Liu et al.'s method already promises low variance, our additional bias reduction allows us to achieve significantly better accuracy for practical problems.

Technically, our new *bias reduction* method provides a new angle of double robustness for off-policy evaluation, orthogonal to the existing literature of doubly robust policy evaluation that solely devotes to *variance reduction* (Jiang & Li, 2016; Thomas & Brunskill, 2016; Farajtabar et al., 2018), mostly based on the idea of control variates (e.g. Asmussen & Glynn, 2007). Our double robustness for bias reduction is significantly different, and instead yields an intriguing connection with the fundamental primal-dual relations between the density (ratio) functions and value functions (e.g., Bertsekas, 2000; Puterman, 2014). This new perspective can inspire more efficient algorithms for policy evaluation, and lead to unified frameworks for these types of double robustness in future work.

## 2 BACKGROUND

**Infinite Horizon Off-Policy Estimation** Let $M = \langle \mathcal{S}, \mathcal{A}, r, \boldsymbol{T}, \mu_0 \rangle$ be a Markov decision process (MDP) with state space $\mathcal{S}$, action space $\mathcal{A}$, reward function $r$, transition probability function $\boldsymbol{T}$, and initial-state distribution $\mu_0$. A policy $\pi$ maps states to distributions over $\mathcal{A}$, with $\pi(a|s)$ being the probability of selecting $a$ given $s$. The average discounted reward for $\pi$, with a given discount $\gamma \in (0, 1)$ [1], is defined as

$$R^\pi := \lim_{T \to \infty} \mathbb{E}_{\tau \sim \pi} \left[ \frac{\sum_{t=0}^T \gamma^t r_t}{\sum_{t=0}^T \gamma^t} \right],$$

where $\tau = \{s_t, a_t, r_t\}_{0 \le t \le T}$ is a trajectory with states, actions, and rewards collected by following policy $\pi$ in the MDP, starting from $s_0 \sim \mu_0$. Given a set of $n$ trajectories, $\mathcal{D} = \{s_t^{(i)}, a_t^{(i)}, r_t^{(i)}\}_{1 \le i \le n, 0 \le t \le T}$, collected under a behavior policy $\pi_0(a|s)$, the off-policy evaluation problem aims to estimate the average discounted reward $R^\pi$ for another target policy $\pi(a|s)$.

**Estimation via Value Function** The value function for policy $\pi$ is defined as the expected accumulated discounted future rewards started from a certain state: $V^\pi(s) = \mathbb{E}_{\tau \sim \pi}[\sum_{t=0}^\infty \gamma^t r_t | s_0 = s]$. We use $r^\pi(s) = \mathbb{E}_{a \sim \pi(\cdot|s)}[r(s, a)]$ to denote the average reward for state $s$ given policy $\pi$. Under the definition, the value function can be seen as a fixed point of the Bellman equation:

$$V^\pi(s) = r^\pi(s) + \gamma \mathcal{P}^\pi V^\pi(s), \quad \mathcal{P}^\pi V^\pi(s) := \mathbb{E}_{a \sim \pi(\cdot|s), s' \sim \boldsymbol{T}(\cdot|s,a)}[V^\pi(s')], \quad \forall s \in \mathcal{S}, \quad (1)$$

where $\mathcal{P}^\pi V(s)$ is the average of the next value function given the current state $s$ and policy $\pi$; see Appendix A.1 for details.

The value function and the expected reward $R^\pi$ is related in the following straightforward way

$$R^\pi = (1 - \gamma) \mathbb{E}_{s \sim \mu_0}[V^\pi(s)], \quad (2)$$

where the expectation is with respect to the distribution $\mu_0(s)$ of the initial states $s_0$ at time $t$. Therefore, given an approximation $\widehat{V}$ of $V^\pi$, and samples $\mathcal{D}_0 := \{s_0^{(i)}\}_{1 \le i \le n_0}$ drawn from $\mu_0(s)$, we can estimate $R^\pi$ by

$$\widehat{R}_{\mathrm{VAL}}^\pi[\widehat{V}] = \frac{(1 - \gamma)}{n_0} \sum_{i=1}^{n_0} \widehat{V}(s_0^{(i)}).$$

Note that this estimator is off-policy in nature, since it requires no samples from the target policy $\pi$.

**Estimation via State Density Function** Denote $d_{\pi,t}(\cdot)$ as average visitation of $s_t$ in time step $t$. The state density function, or the discounted average visitation, is defined as:

$$d_\pi(s) := \lim_{T \to \infty} \frac{\sum_{t=0}^T \gamma^t d_{\pi,t}(s)}{\sum_{t=0}^T \gamma^t} = (1 - \gamma) \sum_{t=0}^\infty \gamma^t d_{\pi,t}(s),$$

---

[1] For average case when $\gamma = 1$, the definition of $R^\pi$ is the same. However, the definition of value function is different. We will assume $\gamma < 1$ throughout our main paper for simplicity; for average case check appendix B for more details.

where $(1 - \gamma)$ can be viewed as the normalization factor introduced by $\sum_{t=0}^{\infty} \gamma^t$.

Similar to Bellman equation for value function, the state density function can also be viewed as a fixed point to the following recursive equation (Liu et al., 2018a, Lemma 3):

$$d_\pi(s') = (1 - \gamma)\mu_0(s') + \gamma \mathcal{T}^\pi d_\pi(s'), \quad \text{where} \quad \mathcal{T}^\pi d_\pi(s') := \sum_{s,a} \boldsymbol{T}(s'|s, a)\pi(a|s)d_\pi(s). \quad (3)$$

The operator $\mathcal{T}^\pi$ is an adjoint operator of $\mathcal{P}^\pi$ used in (1); see Appendix A.1 for a discussion.

If the density function $d_\pi$ is known, it provides an alternative way for estimating the expected reward $R^\pi$, by noting that

$$R^\pi = \mathbb{E}_{s \sim d_\pi, a \sim \pi(\cdot|s)}[r(s, a)]. \quad (4)$$

We can see that both density function $d_\pi$ and value function $V^\pi$ can be used to estimate the expected reward $R^\pi$. We clarify the connection in detail in Appendix A.1.

**Off-Policy State Visitation Importance Sampling** Equation (4) can not be directly used for off-policy estimation, since it involves expectation under the behavior policy $\pi$. Liu et al. (2018a) addressed this problem by introducing a change of measures via importance sampling:

$$R^\pi = \mathbb{E}_{s \sim d_{\pi_0}, a \sim \pi_0(\cdot|s)}\left[w_{\pi/\pi_0}(s)\frac{\pi(a|s)}{\pi_0(a|s)}r(s, a)\right], \quad \text{with} \quad w_{\pi/\pi_0}(s) = \frac{d_\pi(s)}{d_{\pi_0}(s)}, \quad (5)$$

where $w_{\pi/\pi_0}(s)$ is the density ratio function of $d_\pi$ and $d_{\pi_0}$.

Given an approximation $\widehat{w}$ of $w_{\pi/\pi_0}$, and samples $\mathcal{D} = \{s_t^{(i)}, a_t^{(i)}, r_t^{(i)}\}_{1 \leq i \leq n, 0 \leq t \leq T}$ collected from policy $\pi_0$, we can estimate $R^\pi$ as:

$$\widehat{R}_{\text{SIS}}^\pi[\widehat{w}] = \frac{1}{Z}\sum_{i=1}^{n}\sum_{t=0}^{T}\gamma^t \widehat{w}(s_t^{(i)})\frac{\pi(a_i|s_i)}{\pi_0(a_i|s_i)}r_i, \qquad Z = \sum_{i=1}^{n}\sum_{t=0}^{T}\gamma^t \widehat{w}(s_t^{(i)})\frac{\pi(a_t^{(i)}|s_t^{(i)})}{\pi_0(a_t^{(i)}|s_t^{(i)})}, \quad (6)$$

where $Z$ is the normalized constant of the importance weights.

## 3 DOUBLY ROBUST ESTIMATOR

Doubly robust estimator is first proposed into reinforcement learning community to solve contextual bandit problem by Dudík et al. (2011) as an estimator combining *inverse propensity score* (IPS) estimator and *direct method* (DM) estimator.

Jiang & Li (2016) introduce the idea of doubly robust estimator into off-policy evaluation in reinforcement learning. It incorporates an approximate value function as a control variate to reduce the variance of importance sampling estimator. Inspired by previous works, we propose a new doubly robust estimator based on our infinite horizon off-policy estimator $\widehat{R}_{\text{SIS}}^\pi$.

### 3.1 DOUBLY ROBUST ESTIMATOR FOR INFINITE HORIZON MDP

The value-based estimator $\widehat{R}_{\text{VAL}}^\pi[\widehat{V}]$ and density-ratio-based estimator $\widehat{R}_{\text{SIS}}^\pi[\widehat{w}]$ are expected to be accurate when $\widehat{V}$ and $\widehat{w}$ are accurate, respectively. Our goal is to combine their advantages, obtaining a doubly robust estimator that is accurate once either $\widehat{V}$ or $\widehat{w}$ or is accurate.

To simplify the problem, it is useful to examine the limit of infinite samples, with which $\widehat{R}_{\text{VAL}}^\pi[\widehat{V}]$ and $\widehat{R}_{\text{SIS}}^\pi[\widehat{w}]$ converge to the following limit of expectations:

$$R_{\text{SIS}}^\pi[\widehat{w}] := \lim_{n, T \to \infty} \widehat{R}_{\text{SIS}}^\pi[\widehat{w}] = \sum_{s} r^\pi(s)d_{\pi_0}(s)\widehat{w}(s), \quad (7)$$

$$R_{\text{VAL}}^\pi[\widehat{V}] := \lim_{n_0 \to \infty} \widehat{R}_{\text{VAL}}^\pi[\widehat{V}] = (1 - \gamma)\sum_{s} \widehat{V}(s)\mu_0(s). \quad (8)$$

Here and throughout this work, we assume $\widehat{V}$ and $\widehat{w}$ are fixed pre-defined approximations, and only consider the randomness from the data $\mathcal{D}$.

A first observation is that we expect to have $r^\pi \approx \widehat{V} - \gamma \mathcal{P}^\pi \widehat{V}$ by Bellman equation (1), when $\hat{V}$ approximates the true value $V^\pi$. Plugging this into $R_{\text{SIS}}^\pi[\widehat{w}]$ in Equation (7), we obtain the following "bridge estimator", which incorporates information from both $\widehat{w}$ and $\widehat{V}$:

$$R_{\text{bridge}}^\pi[\widehat{V}, \widehat{w}] = \sum_s \left( \widehat{V}(s) - \gamma \mathcal{P}^\pi \widehat{V}(s) \right) d_{\pi_0}(s) \widehat{w}(s), \tag{9}$$

where operator $\mathcal{P}^\pi$ is defined in Bellman equation (1). The corresponding empirical estimator is defined by

$$\widehat{R}_{\text{bridge}}^\pi[\widehat{V}, \widehat{w}] = \sum_{i=1}^n \sum_{t=0}^{T-1} \left( \frac{1}{Z_1} \gamma^t \widehat{w}(s_t^{(i)}) \widehat{V}(s_t^{(i)}) - \frac{1}{Z_2} \gamma^{t+1} \widehat{w}(s_t^{(i)}) \frac{\pi(a_i|s_i)}{\pi_0(a_i|s_i)} \widehat{V}(s_{t+1}^{(i)}) \right), \tag{10}$$

where $Z_1 = \sum_{i=1}^n \sum_{t=0}^{T-1} \gamma^t \widehat{w}(s_t^{(i)})$ and $Z_2 = \sum_{i=1}^n \sum_{t=0}^{T-1} \gamma^{t+1} \widehat{w}(s_t^{(i)}) \beta_{\pi/\pi_0}(a_t^{(i)}|s_t^{(i)})$ are self-normalized constant of important weights each empirical estimation.

However, directly estimating $R^\pi$ using the bridge estimator $\widehat{R}_{\text{bridge}}^\pi[\widehat{V}, \widehat{w}]$ yields a poor estimation, because it includes the errors from both $\widehat{w}$ and $\widehat{V}$ and can be "*doubly worse*". However, we can construct our "*doubly robust*" estimator by canceling $R_{\text{bridge}}^\pi[\widehat{V}, \widehat{w}]$ out from $R_{\text{SIS}}^\pi[\widehat{w}]$ and $R_{\text{VAL}}^\pi[\widehat{V}]$:

$$R_{\text{DR}}^\pi[\widehat{V}, \widehat{w}] = \underbrace{\sum_s r^\pi(s) d_{\pi_0}(s) \widehat{w}(s)}_{R_{\text{SIS}}^\pi[\widehat{w}]} + \underbrace{(1-\gamma) \sum_s \widehat{V}(s) \mu_0(s)}_{R_{\text{VAL}}^\pi[\widehat{V}]} - \underbrace{\sum_s \left( \widehat{V}(s) - \gamma \mathcal{P}^\pi \widehat{V}(s) \right) d_{\pi_0}(s) \widehat{w}(s)}_{R_{\text{bridge}}^\pi[\widehat{V}, \widehat{w}]}. \tag{11}$$

And its corresponding empirical estimator can be written as:

$$\widehat{R}_{\text{DR}}^\pi[\widehat{V}, \widehat{w}] = \widehat{R}_{\text{SIS}}^\pi[\widehat{w}] + \widehat{R}_{\text{VAL}}^\pi[\widehat{V}] - \widehat{R}_{\text{bridge}}^\pi[\widehat{V}, \widehat{w}].$$

**Doubly Robust Bias Reduction**  The double robustness of $\widehat{R}_{\text{DR}}^\pi[\widehat{V}, \widehat{w}]$ is reflected in the following key theorem, which shows that it is accurate once either $\widehat{V}$ or $\widehat{w}$ is accurate.

**Theorem 3.1** (Doubly Robustness). *Let $R_{DR}^\pi[\widehat{V}, \widehat{w}] := \lim_{n_0, n, T \to \infty} \widehat{R}_{DR}^\pi[\widehat{V}, \widehat{w}]$ be the limit of $\widehat{R}_{DR}^\pi$ when it has infinite samples. Following the definition above, we have*

$$R_{DR}^\pi[\widehat{V}, \widehat{w}] - R^\pi = \mathbb{E}_{s \sim d_{\pi_0}} \left[ \varepsilon_{\widehat{w}}(s) \varepsilon_{\widehat{V}}(s) \right], \tag{12}$$

*where $\varepsilon_{\widehat{V}}$ and $\varepsilon_{\widehat{w}}$ are errors of $\widehat{V}$ and $\widehat{w}$, respective, defined as follows*

$$\varepsilon_{\widehat{w}} = \frac{d_\pi(s)}{d_{\pi_0}(s)} - \widehat{w}(s), \qquad\qquad \varepsilon_{\widehat{V}}(s) = \widehat{V}(s) - r^\pi(s) - \gamma \mathcal{P}^\pi \widehat{V}(s).$$

*The error $\varepsilon_{\widehat{w}}$ of $\widehat{w}$ is measured by the difference with the true density ratio $d_\pi(s)/d_{\pi_0}(s)$, and the error $\varepsilon_{\widehat{V}}$ of $\widehat{V}$ is measured using the Bellman residual.*

From the theorem we can see that, if $\widehat{V}$ is exact ($\widehat{V} \equiv V^\pi$), we have $\varepsilon_{\widehat{V}} \equiv 0$; if $\widehat{w}$ is exact ($\widehat{w} \equiv d_\pi/d_{\pi_0}$), we have $\varepsilon_{\widehat{w}} \equiv 0$. Therefore, our estimator is consistent (i.e., $\lim_{n, n_0 \to \infty} \widehat{R}_{\text{DR}}^\pi[\widehat{V}, \widehat{w}] = R^\pi$) if either $\widehat{V}$ or $\widehat{w}$ are exact. The estimator is thus doubly robust in this sense. In contrast, $\widehat{R}_{\text{SIS}}^\pi[\widehat{w}]$ and $\widehat{R}_{\text{VAL}}^\pi[\widehat{V}]$ can be more sensitive to the error of $\widehat{w}$ and $\widehat{V}$, respectively:

$$R_{\text{SIS}}^\pi[\widehat{w}] - R^\pi = \mathbb{E}_{s \sim d_{\pi_0}} \left[ \varepsilon_{\widehat{w}}(s) r^\pi(s) \right], \qquad R_{\text{VAL}}^\pi[\widehat{V}] - R^\pi = \mathbb{E}_{s \sim d_{\pi_0}} \left[ w_{\pi/\pi_0}(s) \varepsilon_{\widehat{V}}(s) \right].$$

**Variance Analysis**  Different from the bias reduction, the doubly robust estimator does not guarantee to the reduce the variance over $\widehat{R}_{\text{SIS}}^\pi[\widehat{w}]$ or $\widehat{R}_{\text{VAL}}^\pi[\widehat{V}]$ in general. However, as we show in the following result, we can break the variance of $\widehat{R}_{\text{DR}}^\pi[\widehat{V}, \widehat{w}]$ into two parts. The first is the variance of $\widehat{R}_{\text{VAL}}^\pi[\widehat{V}]$ which is often relatively small. The second is generally no greater than the variance of $\widehat{R}_{\text{SIS}}^\pi[\widehat{w}]$, and can be much smaller when $\widehat{V} \approx V^\pi$. Moreover, $\widehat{R}_{\text{VAL}}^\pi[\widehat{V}]$ and $\widehat{R}_{\text{SIS}}^\pi[\widehat{w}]$ avoid the curse of horizon by design, so their variances tend to be much smaller than the corresponding estimators that apply IPS correction on the trajectories.

---

**Algorithm 1** Infinite Horizon Doubly Robust Estimator

---

**Input**: Transition data $\mathcal{D}_{\pi_0} = \{s_t^{(i)}, a_t^{(i)}, r_t^{(i)}\}_{1 \le i \le n, 0 \le t \le T}$ from policy $\pi_0$;
$\qquad \mathcal{D}_0 = \{s_0^{(j)}\}_{1 \le j \le n_0}$ be samples from initial distribution $\mu_0$
$\qquad$ target policy $\pi$, value function estimate $\widehat{V}$; density ratio estimate $\widehat{w}$.
**Estimation:** Use $\widehat{R}_{\text{DR}}^{\pi}$ in (11) to estimate $R^{\pi}$ using sample from $\mathcal{D}$ and $\mathcal{D}_0$.

---

**Proposition 3.1** (Variance Analysis). *Assume $\widehat{R}_{DR}^{\pi}[\widehat{V}, \widehat{w}]$ is estimated based on sample $\mathcal{D}_0 \sim \mu_0$ and $\mathcal{D}_{\pi_0} \sim d_{\pi_0}$, which we assume to be independent of each other. We have*

$$Var_{\mathcal{D}_0, \mathcal{D}_{\pi_0}}\left[\widehat{R}_{DR}^{\pi}[\widehat{V}, \widehat{w}]\right] = Var_{\mathcal{D}_0}\left[\widehat{R}_{VAL}^{\pi}[\widehat{V}]\right] + Var_{\mathcal{D}_{\pi_0}}\left[\widehat{R}_{res}^{\pi}[\widehat{V}, \widehat{w}]\right], \qquad (13)$$

*with $\widehat{R}_{res}^{\pi}[\widehat{V}, \widehat{w}] := \widehat{R}_{SIS}^{\pi}[\widehat{w}] - \widehat{R}_{bridge}^{\pi}[\widehat{V}, \widehat{w}] = \widehat{\mathbb{E}}_{\mathcal{D}_{\pi_0}}\left[\widehat{\varepsilon}_{\widehat{V}}(s)\widehat{w}(s)\right]$ and $\widehat{\varepsilon}_{\widehat{V}}(s)$ the TD error of $\widehat{V}$,*

$$\widehat{\varepsilon}_{\widehat{V}}(s) = \widehat{r^{\pi}}(s) - \widehat{V}(s) + \gamma\widehat{\mathcal{P}^{\pi}}\widehat{V}(s),$$

*with $\widehat{r^{\pi}}(s) = r(s,a)\pi(a|s)/\pi_0(a|s)$, $\widehat{\mathcal{P}^{\pi}}\widehat{V}(s) = \pi(a|s)/\pi_0(a|s)\widehat{V}(s')$. Therefore, $Var_{\mathcal{D}_{\pi_0}}(\widehat{R}_{res}^{\pi}[\widehat{V}, \widehat{w}])$ can be small when $\widehat{V}$ is close to the true value $V^{\pi}$, or $\widehat{\varepsilon}_{\widehat{V}}(s) \approx 0$.*

From the proposition we can see that when $\widehat{V}$ is close to the true value $V^{\pi}$, the variance of the residual $\widehat{\varepsilon}_{\widehat{V}}$ may be negligibly small compared to the variance of $\widehat{R}_{\text{SIS}}^{\pi}[\widehat{w}]$. A further comparison of the variances between $\widehat{R}_{\text{res}}^{\pi}[\widehat{V}, \widehat{w}]$ and $\widehat{R}_{\text{SIS}}^{\pi}[\widehat{w}]$ is provided in Appendix A.3. In the case when the TD error $\widehat{\varepsilon}_{\widehat{V}}$ is negligible, we have $Var_{\mathcal{D}_0, \mathcal{D}_{\pi_0}}\left[\widehat{R}_{\text{DR}}^{\pi}[\widehat{V}, \widehat{w}]\right] \approx Var_{\mathcal{D}_0}\left[\widehat{R}_{\text{VAL}}^{\pi}[\widehat{V}]\right]$. Typically, the variance of $\widehat{R}_{\text{VAL}}^{\pi}[\widehat{V}]$ is smaller than $\widehat{R}_{\text{SIS}}^{\pi}[\widehat{w}]$ since the variance of importance sampling methods heavily depends on the effective sample size, which is less controllable compared to $\widehat{R}_{\text{VAL}}^{\pi}[\widehat{V}]$. Therefore, the variance of our doubly robust estimator may even be smaller than that of $\widehat{R}_{\text{SIS}}^{\pi}[\widehat{w}]$ in practice.

The variance in (13) is a sum of two terms because of the assumption that samples from $\mu_0$ and $d_{\pi_0}$ are independent. In practice, they have dependency but it is possible to couple the samples from $\mu_0$ and $d_{\pi_0}$ in a certain way to further decrease the variance, which we leave as future work.

**Proposed Algorithm for Off-Policy Evaluation** Suppose we have already obtained $\widehat{V}$ and $\widehat{w}$, as estimations of $V^{\pi}$ and $w_{\pi/\pi_0}$, respectively, we can directly use equation (11) to estimate $R^{\pi}$. A detail procedure is described in Algorithm 1.

## 4 DOUBLE ROBUSTNESS AND LAGRANGIAN DUALITY

In this section, we reveal a surprising connection between our double robustness and Lagrangian duality. We show that our doubly robust estimator is equivalent to the Lagrangian function of a primal-dual formulation of policy evaluation. This connection is of interest in itself, and may provide a foundation for deriving more effective algorithms.

We start with the following classic, optimization formulation of policy evaluation (Puterman, 2014):

$$R^{\pi} = \min_{V}\left\{(1-\gamma)\sum_s \mu_0(s)V(s) \quad \text{s.t.} \quad V(s) \ge r^{\pi}(s) + \gamma\mathcal{P}^{\pi}V(s), \quad \forall s\right\}, \qquad (14)$$

where we find $V$ to maximize its average value, subject to an inequality constraint on the Bellman equation. It can be shown that the solution of (14) is achieved by the true value function $V^{\pi}$, hence yielding a true expected reward $R^{\pi}$.

Introducing a Lagrangian multiplier $\rho \ge 0$, we can derive the Lagrangian function $L(V, \rho)$ of (14),

$$L(V, \rho) = (1-\gamma)\sum_s \mu_0(s)V(s) - \sum_s \rho(s)(V(s) - r^{\pi}(s) + \gamma\mathcal{P}^{\pi}V(s)). \qquad (15)$$

Comparing $L(V, \rho)$ with our estimator $\widehat{R}_{\text{DR}}^{\pi}[\widehat{V}, \widehat{w}]$ in (11), we can see that they are in fact equivalent in expectation.

**Theorem 4.1** (Primal Dual). *I) Define $w_{\rho/\pi_0}(s) = \frac{\rho(s)}{d_{\pi_0}(s)}$. We have*

$$L(V, \rho) = R_{DR}^\pi[V, w_{\rho/\pi_0}], \qquad \text{and hence} \qquad L(V, d_\pi) = L(V^\pi, \rho) = R^\pi, \ \forall \, V, \rho,$$

*which suggests that $L(V, \rho)$ is "doubly robust" in that it equals $R^\pi$ if either $V = V^\pi$ or $\rho = d_\pi$.*

*II) The primal problem (14) forms a strong duality with the following dual problem,*

$$R^\pi = \max_{\rho \geq 0} \left\{ \sum_s \rho(s) r^\pi(s) \qquad s.t. \qquad \rho(s') = (1 - \gamma)\mu_0(s') + \gamma \mathcal{T}^\pi \rho(s'), \quad \forall s' \right\}, \qquad (16)$$

*where $\mathcal{T}^\pi$ is defined in (3).*

This shows that the dual problem is equivalent to constraint $\rho$ using the fixed point equation (3) and maximize the average reward given distribution $\rho$. Since the unique fixed point of (3) is $d_\pi(s)$, the solution of (16) naturally yields the true reward $R^\pi$, hence forming a zero duality gap with (14).

It is natural to intuit the double robustness of the Lagrangian function. From (15), $L(V, \rho)$ can be viewed as estimating the reward $R^\pi$ using value function with a correction of Bellman residual $(V - r^\pi - \gamma \mathcal{P}^\pi V)$. If $V = V^\pi$, the estimation equals the true reward and the correction equals zero. From the dual problem (16), $L(V, \rho)$ can be viewed as estimating $R^\pi$ using density function $\rho$, corrected by the residual $(\rho - (1 - \gamma)\mu_0 - \gamma \mathcal{T}^\pi \rho)$. We again get the true reward if $\rho = d_\pi$.

It turns out that we can use the primal-dual formula when $\gamma = 1$ to obtain the double robust estimator for the average reward case. We clarify it in appendix B.

**Remark**   The fact that the density function $d_\pi$ forms a dual variable of the value function $V^\pi$ is widely known in the optimal control and reinforcement learning literature (e.g., Bertsekas, 2000; Puterman, 2014; de Farias & Van Roy, 2003), and has been leveraged in various works for policy optimization. However, it does not seem to be well exploited in the literature of off-policy policy evaluation.

## 5 RELATED WORK

**Off-Policy Value Evaluation**   The problem of off-policy value evaluation has been studied extensively in contextual bandits and MDPs (Fonteneau et al., 2013; Li et al., 2015; Jiang & Li, 2016; Thomas & Brunskill, 2016; Liu et al., 2018b; Farajtabar et al., 2018; Hanna et al., 2019; Xie et al., 2019; Rowland et al., 2019). However, most of the existing works are based on importance sampling (IS) to correct the mismatch between the distribution of the whole trajectories induced by the behavior and target policies, which faces the "curse of horizon" (Liu et al., 2018a) when extended to long-horizon (or infinite-horizon) problems.

Several other works (Guo et al., 2017; Hallak & Mannor, 2017; Liu et al., 2018a; Gelada & Bellemare, 2019; Nachum et al., 2019; Liu et al., 2019a) have been proposed to address the high variance issue in the long-horizon problems. Liu et al. (2018a) apply importance sampling on the average visitation distribution of state-action pairs, instead of the distribution of the whole trajectories, providing an effective approach to break "the curse of horizon". However, they require to learn a density ratio function over the whole state-action pairs, which may induce large bias. Our work incorporates the density ratio and value function estimation, which significantly reduces the induced bias of two estimators, resulting a doubly robust estimator.

Our work is also closely related to DR techniques used in finite horizon problems (Murphy et al., 2001; Dudík et al., 2011; Jiang & Li, 2016; Thomas & Brunskill, 2016; Farajtabar et al., 2018), which incorporate an approximate value function as control variates to IS estimators. Different from existing DR approaches, our work is related to the well known duality between the density and the value function, which reveals the relationship between density (ratio) learning (Liu et al., 2018a) and value function learning. Based on this interesting observation, we further obtain the doubly robust estimator for estimating average reward in infinite-horizon problems.

A recent work (Xie et al., 2019) has also explored the idea of tracking marginal state distribution shifts at every single time step in an episode, instead of the stationary state distribution across all time steps (Liu et al., 2018a). It provides additional benefits for understanding and analyzing problems

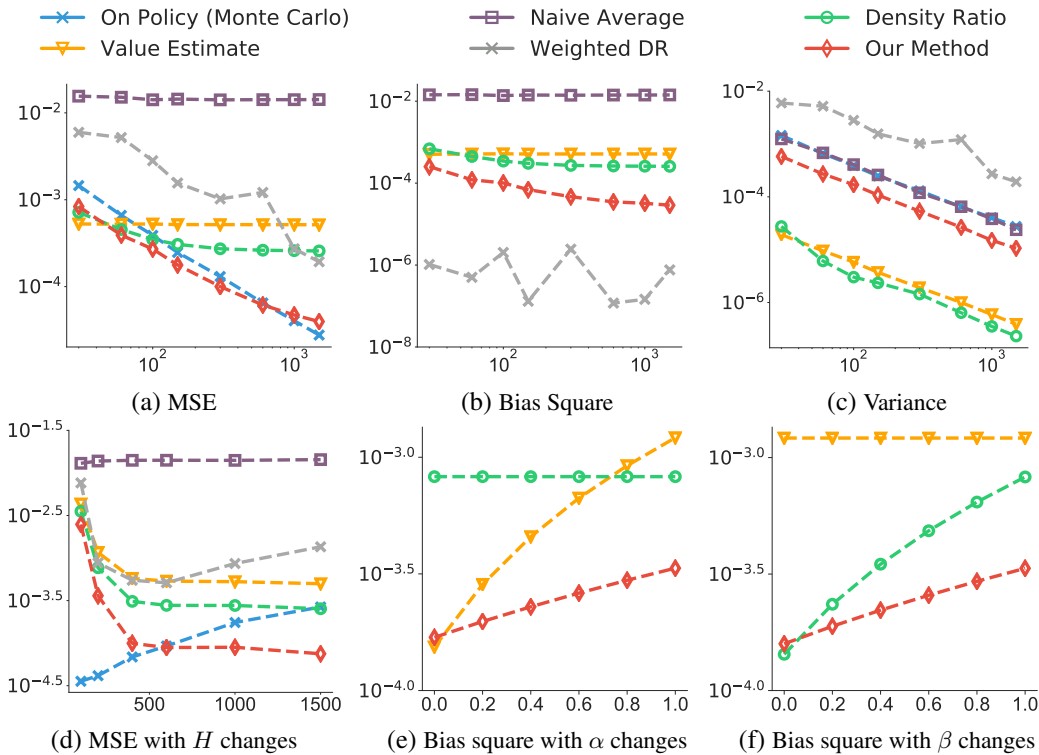

Figure 1: Off Policy Evaluation Results on Taxi. Default parameter, discounted factor $\gamma = 0.99$, mixed ratio $\alpha = \beta = 1$, horizon length $H = 600$. For (a)-(c) the x-axis is the number of trajectories and y-axis corresponds to MSE, Bias Square and Variance, respectively. For (d) we fix the total number of samples (number of trajectories times horizon length) and change the horizon length as x-axis and observe the MSE. (e) and (f) show the change the mixed ratio of $\alpha$, $\beta$ with the change of bias. We repeat each experiment for 1000 runs.

such as short horizon and time-variant MDPs. They call this approach *marginalized importance sampling* (MIS). Later on, Kallus & Uehara (2019a) incorporate the DR technique to improve the MIS estimator. Independent of this work, Kallus & Uehara (2019b); Uehara et al. (2019) extend previous works to propose estimators can be viewed as a variant of ours.

**Primal-Dual Value Learning**   Primal-dual optimization techniques have been widely used for off-policy value function learning and policy optimization (Liu et al., 2015; Chen & Wang, 2016; Dai et al., 2017; 2018; Feng et al., 2019). Nevertheless, the duality between density and value function has not been well explored in the literature of off policy value estimation. Our work proposes a new doubly robustness technique for off-policy value estimation, which can be naturally viewed as the Lagrangian function of the primal-dual formulation of policy evaluation, providing an alternative unified view for off policy value evaluation.

## 6 EXPERIMENT

In this section, we conduct simulation experiments on different environmental settings to compare our new doubly robust estimator with existing methods. We mainly compare with infinite horizon based estimator including state importance sampling estimator (Liu et al. (2018a)) and value function estimator. We do not report results on the vanilla trajectory-based importance sampling estimators because of their significant higher variance, but we do compare with the doubly robust version induced by Thomas & Brunskill (2016) (self-normalized variant of Jiang & Li (2016)). In all experiments we compare with Monte Carlo and naive average as Liu et al. (2018a) suggested. The ground truth for each environment is calculated by averaging Monte Carlo estimation with a very large sample size.

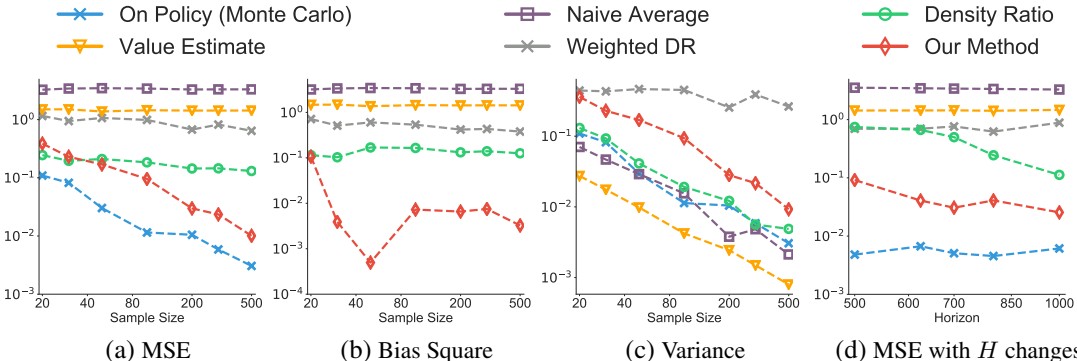

Figure 2: Off Policy Evaluation Results on Puck-Mountain. We set discounted factor $\gamma = 0.995$ as default. For (a)-(c) we set the horizon $H = 1000$ and the x-axis is the number of trajectories for used for evaluation. For (d) we fix the total number of samples and change the horizon length.

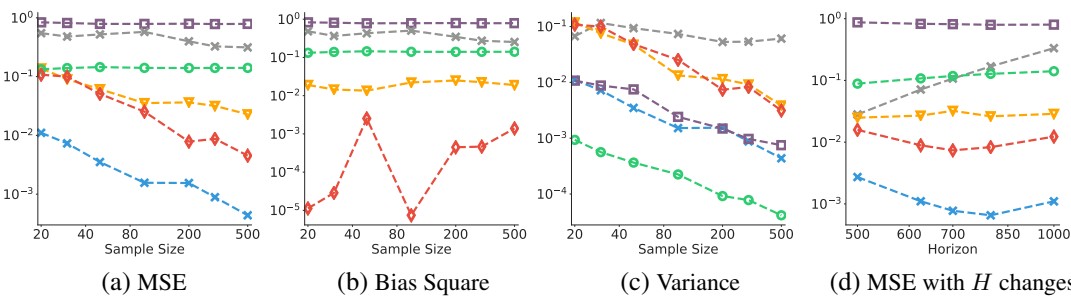

Figure 3: Off Policy Evaluation Results on InvertedPendulum-v2. We set discounted factor $\gamma = 0.995$ as default. For (a)-(c) we set the horizon $H = 1000$ and the x-axis is the number of trajectories for used for evaluation. For (d) we fix the total number of samples and change the horizon length.

**Taxi Environment**    We follow Liu et al. (2018a)'s tabular environment *Taxi*, which has 2000 states and 6 actions in total. For more experimental details, please check appendix C.1.

We pre-train two different $\widehat{V}$ and $\tilde{V}$ trained with a small and fairly large size of samples, respectively, where $\tilde{V}$ is very close to true value function $V^{\pi}$ but $\widehat{V}$ is relatively further from it. Similarly we pre-train $\widehat{\rho}$ and $\tilde{\rho} \approx d_{\pi}$. For estimation we use a mixed ratio $\alpha, \beta$ to control the bias of the input $V, \rho$, where $V = \alpha\widehat{V} + (1-\alpha)\tilde{V}$ and $\rho = \beta\widehat{\rho} + (1-\beta)\tilde{\rho}$.

Figure 1(a)-(c) show results of comparison for different methods as we changing the number of trajectories. We can see that the MSE performance of value function($\widehat{R}^{\pi}_{\text{VAL}}$) and state visitation importance sampling($\widehat{R}^{\pi}_{\text{SIS}}$) estimators are mainly impeded by their large biases, while our method has much less bias thus it can keep decreasing as sample size increase and achieves same performance as on policy estimator. Figure 1(d) shows results if we change the horizon length. Notice that here we keep the number of samples to be the same, so if we increase our horizon length we will decrease the number of trajectories in the same time. We can see that our method alongside with all infinite horizon methods will get better result as horizon length increase. Figure 1(e)-(f) indicate the "double robustness" of our method, where our method benefits from either a better $V$ or a better $\rho$.

**Puck-Mountain**    Puck-Mountain is an environment similar to Mountain-Car, except that the goal of Puck-Mountain is to push the puck as high as possible in a local valley, which has a continuous state space of $\mathbb{R}^2$ and a discrete action space similar to Mountain-Car. We use the softmax functions of an optimal Q-function as both target policy and behavior policy, where the temperature of the behavior policy is higher (encouraging exploration). For more details of constructing policies and training algorithms for density ratio and value functions, please check appendix C.2.

Figure 2(a)-(c) show results of comparison for different methods as we changing the number of trajectories. Similar to taxi, we find our method has much lower bias than density ratio and value function estimation, which yields a better MSE. In Figure 2(d) the performance for all infinite horizon estimator will not degenerate as horizon increases, while finite horizon method such as finite weighted horizon doubly robust will suffer from larger variance as horizon increases.

**InvertedPendulum**    InvertedPendulum is a pendulum that has its center of mass above its pivot point. We use the implementation of InvertedPendulum from OpenAI gym (Brockman et al., 2016), which is a continuous control task with state space in $\mathcal{R}^4$ and we discrete the action space to be $\{-1, -0.3, -0.2, 0, 0.2, 0.3, 1\}$. More experiment details can be found in appendix C.2.

In Figure 3(a)-(c) our method again significantly reduces the bias, which yields a better MSE comparing with value and density estimation. Figure 3(d) also shows that our method consistently outperforms all other methods as the horizon increases with a fixed total timesteps.

## 7    CONCLUSION

In this paper, we develop a new doubly robust estimator based on the infinite horizon density ratio and off policy value estimation. Our new proposed doubly robust estimator can be accurate as long as one of the estimators are accurate, which yields a significant advantage comparing to previous estimators. Future directions include deriving more novel optimization algorithms to learn value function and density(ratio) function by using the primal dual framework.

## ACKNOWLEDGEMENT

This work is supported in part by NSF CRII 1830161 and NSF CAREER 1846421. We would like to acknowledge Google Cloud for their support.

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

# A PROOF

## A.1 TRANSITION OPERATOR FOR BELLMAN EQUATION

For simplicity, we define the following two operators thorough our proofs to simplify our notations.

**Definition A.1.** *Given a policy $\pi$ and the unknown environment transition $\boldsymbol{T}$, we define $\mathcal{T}^\pi$ and $\mathcal{P}^\pi$ over any function $f : \mathcal{S} \to \mathcal{R}$ as*

$$(\mathcal{T}^\pi f)(s') = \sum_{s,a} \boldsymbol{T}(s'|s,a)\pi(a|s)f(s)$$

$$(\mathcal{P}^\pi f)(s) = \sum_{s',a} \boldsymbol{T}(s'|s,a)\pi(a|s)f(s')$$

Using these operator notations, we can rewrite the above two recursive equations as:

$$V^\pi = r^\pi + \gamma \mathcal{P}^\pi V^\pi,$$
$$d_\pi = (1-\gamma)\mu_0 + \gamma \mathcal{T}^\pi d_\pi,$$

where $r^\pi(s) = \mathbb{E}_{a\sim\pi(\cdot|s)}[r(s,a)]$.

These transition operators have the following nice adjoint property.

**Lemma A.2.** *For two function $f$ and $g$, if the following summation is finite, we will have*

$$\sum_s (\mathcal{P}^\pi f)(s)g(s) = \sum_s f(s)(\mathcal{T}^\pi g)(s). \tag{17}$$

*Proof.*

$$\sum_s (\mathcal{P}^\pi f)(s)g(s) = \sum_s \left(\sum_{s',a} \boldsymbol{T}(s'|s,a)\pi(a|s)f(s')\right) g(s)$$

$$= \sum_{s'} f(s')\left(\sum_{s,a} \boldsymbol{T}(s'|s,a)\pi(a|s)g(s)\right)$$

$$= \sum_{s'} f(s')(\mathcal{T}^\pi g)(s')$$

$\square$

Using this property we can actually using Bellman Equations to re-derive the two different way to get $R^\pi$.

$$R^\pi = \lim_{T\to\infty} \mathbb{E}_{\tau^{(i)}\sim\pi}\left[\frac{\sum_{t=0}^T \gamma^t r_t}{\sum_{t=0}^T \gamma^t}\right]$$

$$= \sum_s V^\pi(s)(1-\gamma)\mu_0(s)$$

$$= \sum_s V^\pi(s)(\mathcal{I}-\gamma\mathcal{T}^\pi)d_\pi(s)$$

$$= \sum_s (\mathcal{I}-\gamma\mathcal{P}^\pi)V^\pi(s)d_\pi(s)$$

$$= \sum_s r^\pi(s)d_\pi(s).$$

## A.2 PROOF OF THEOREM 3.1

**Theorem 3.1** (Doubly Robustness). *Let $R_{DR}^\pi[\widehat{V}, \widehat{w}] := \lim_{n_0, n, T \to \infty} \widehat{R}_{DR}^\pi[\widehat{V}, \widehat{w}]$ be the limit of $\widehat{R}_{DR}^\pi$ when it has infinite samples. Following the definition above, we have*

$$R_{DR}^\pi[\widehat{V}, \widehat{w}] - R^\pi = \mathbb{E}_{s \sim d_{\pi_0}}\left[\varepsilon_{\widehat{w}}(s)\varepsilon_{\widehat{V}}(s)\right], \tag{12}$$

*where $\varepsilon_{\widehat{V}}$ and $\varepsilon_{\widehat{w}}$ are errors of $\widehat{V}$ and $\widehat{w}$, respective, defined as follows*

$$\varepsilon_{\widehat{w}} = \frac{d_\pi(s)}{d_{\pi_0}(s)} - \widehat{w}(s), \qquad \varepsilon_{\widehat{V}}(s) = \widehat{V}(s) - r^\pi(s) - \gamma\mathcal{P}^\pi\widehat{V}(s).$$

*The error $\varepsilon_{\widehat{w}}$ of $\widehat{w}$ is measured by the difference with the true density ratio $d_\pi(s)/d_{\pi_0}(s)$, and the error $\varepsilon_{\widehat{V}}$ of $\widehat{V}$ is measured using the Bellman residual.*

*Proof.* Using the property of the operator, we can rewrite $(1 - \gamma)\mu_0(s)$ using Bellman equation as $d_\pi - \gamma\mathcal{T}^\pi d_\pi$, thus we have

$$\begin{aligned}
R_{\text{VAL}}^\pi[\widehat{V}] &= (1 - \gamma)\sum_s \widehat{V}(s)\mu_0(s) \\
&= \sum_s \widehat{V}(s)\left(d_\pi - \gamma\mathcal{T}^\pi d_\pi\right)(s) \\
&= \sum_s \left(I - \gamma\mathcal{P}^\pi\right)\widehat{V}(s)d_\pi(s).
\end{aligned}$$

and similarly if we break $r^\pi$ as $(I - \gamma\mathcal{P}^\pi)V^\pi$, for $R_{\text{SIS}}^\pi[\widehat{w}]$ we have:

$$R_{\text{SIS}}^\pi[\widehat{w}] = \sum_s \left(I - \gamma\mathcal{P}^\pi\right)V^\pi(s)d_{\widehat{w}}(s),$$

where $d_{\widehat{w}} = d_{\pi_0}\widehat{w}$ for short.

Compare with $R^\pi = \sum_s (I - \gamma\mathcal{P}^\pi)V^\pi(s)d_\pi(s)$, we can see the main difference between $R_{\text{SIS}}^\pi$ and $R_{\text{VAL}}^\pi$ with $R^\pi$ are they replace $d_\pi$ and $d_{\widehat{w}}$ and $V^\pi$ with $\widehat{V}$ respectively. If we add them together and minus the connection estimator, we have we will have:

$$\begin{aligned}
R_{\text{DR}}^\pi[\widehat{V}, \widehat{w}] - R^\pi &= R_{\text{SIS}}^\pi[\widehat{w}] + R_{\text{VAL}}^\pi[\widehat{V}] - R_{\text{bridge}}^\pi[\widehat{V}, \widehat{w}] - R^\pi \\
&= \sum_s \left(\left(I - \gamma\mathcal{P}^\pi\right)V^\pi(s)d_{\widehat{w}}(s) + \left(I - \gamma\mathcal{P}^\pi\right)\widehat{V}(s)d_\pi(s)\right. \\
&\qquad\qquad \left. - \left(I - \gamma\mathcal{P}^\pi\right)\widehat{V}(s)d_{\widehat{w}}(s) - \left(I - \gamma\mathcal{P}^\pi\right)V^\pi(s)d_\pi(s)\right) \\
&= \sum_s \left(I - \gamma\mathcal{P}^\pi\right)\left(V^\pi - \widehat{V}\right)(s)(d_{\widehat{w}} - d_\pi)(s) \\
&= \sum_s d_{\pi_0}(s)\varepsilon_{\widehat{V}}(s)\varepsilon_{\widehat{w}}(s) \\
&= \mathbb{E}_{s \sim d_{\pi_0}}\left[\varepsilon_{\widehat{V}}(s)\varepsilon_{\widehat{w}}(s)\right],
\end{aligned}$$

where the third equation is because $(I - \gamma\mathcal{P}^\pi)(V^\pi - \widehat{V}) = (I - \gamma\mathcal{P}^\pi)V^\pi - (I - \gamma\mathcal{P}^\pi)\widehat{V} = r^\pi - (I - \gamma\mathcal{P}^\pi)\widehat{V}$. Similarly, the bias for $R_{\text{DR}}^\pi$ and $R_{\text{SIS}}^\pi$ can be calculated as:

$$\begin{aligned}
R_{\text{SIS}}^\pi[\widehat{w}] - R^\pi &= \sum_s \left(I - \gamma\mathcal{P}^\pi\right)V^\pi(s)\left(d_{\widehat{w}}(s) - d_\pi(s)\right) \\
&= \sum_s r^\pi(s)d_{\pi_0}(s)\varepsilon_{\widehat{w}}(s) \\
&= \mathbb{E}_{s \sim d_{\pi_0}}\left[\varepsilon_{\widehat{w}}(s)r^\pi(s)\right],
\end{aligned}$$

and

$$\begin{aligned}
R_{\text{VAL}}^\pi[\widehat{V}] - R^\pi &= \sum_s \left(I - \gamma\mathcal{P}^\pi\right)\left(\widehat{V}(s) - V^\pi(s)\right)d_\pi(s) \\
&= \sum_s \varepsilon_{\widehat{V}}(s)w_{\pi/\pi_0}(s)d_{\pi_0}(s) \\
&= \mathbb{E}_{s \sim d_{\pi_0}}\left[w_{\pi/\pi_0}(s)\varepsilon_{\widehat{V}}(s)\right].
\end{aligned}$$

$\square$

A.3   MORE DISCUSSIONS ON THE VARIANCE IN PROPOSITION 3.1

Recall the proposition 3.1.

**Proposition 3.1** (Variance Analysis). *Assume* $\widehat{R}_{DR}^{\pi}[\widehat{V}, \widehat{w}]$ *is estimated based on sample* $\mathcal{D}_0 \sim \mu_0$ *and* $\mathcal{D}_{\pi_0} \sim d_{\pi_0}$, *which we assume to be independent of each other. We have*

$$Var_{\mathcal{D}_0, \mathcal{D}_{\pi_0}} \left[ \widehat{R}_{DR}^{\pi}[\widehat{V}, \widehat{w}] \right] = Var_{\mathcal{D}_0} \left[ \widehat{R}_{VAL}^{\pi}[\widehat{V}] \right] + Var_{\mathcal{D}_{\pi_0}} \left[ \widehat{R}_{res}^{\pi}[\widehat{V}, \widehat{w}] \right], \qquad (13)$$

*with* $\widehat{R}_{res}^{\pi}[\widehat{V}, \widehat{w}] := \widehat{R}_{SIS}^{\pi}[\widehat{w}] - \widehat{R}_{bridge}^{\pi}[\widehat{V}, \widehat{w}] = \widehat{\mathbb{E}}_{\mathcal{D}_{\pi_0}} \left[ \widehat{\varepsilon}_{\widehat{V}}(s) \widehat{w}(s) \right]$ *and* $\widehat{\varepsilon}_{\widehat{V}}(s)$ *the TD error of* $\widehat{V}$,

$$\widehat{\varepsilon}_{\widehat{V}}(s) = \widehat{r^{\pi}}(s) - \widehat{V}(s) + \gamma \widehat{\mathcal{P}^{\pi} \widehat{V}}(s),$$

*with* $\widehat{r^{\pi}}(s) = r(s,a)\pi(a|s)/\pi_0(a|s)$, $\widehat{\mathcal{P}^{\pi}\widehat{V}}(s) = \pi(a|s)/\pi_0(a|s)\widehat{V}(s')$. *Therefore,* $Var_{\mathcal{D}_{\pi_0}}(\widehat{R}_{res}^{\pi}[\widehat{V}, \widehat{w}])$ *can be small when* $\widehat{V}$ *is close to the true value* $V^{\pi}$, *or* $\widehat{\varepsilon}_{\widehat{V}}(s) \approx 0$.

*Proof.* Since $\widehat{R}_{DR}^{\pi}[\widehat{V}, \widehat{w}] = \widehat{R}_{VAL}^{\pi}[\widehat{V}] + \widehat{R}_{res}^{\pi}[\widehat{V}, \widehat{w}]$ and we have the assumption that the samples from $\mathcal{D}_0$ and $\mathcal{D}_{\pi_0}$ are independent, the proposition comes immediately. $\square$

Intuitively, the variance of $\widehat{R}_{res}^{\pi}[\widehat{V}, \widehat{w}]$ is less than $\widehat{R}_{SIS}^{\pi}[\widehat{w}]$. We want to quantitatively study every pieces of orthogonal randomness of $\widehat{R}_{res}^{\pi}[\widehat{V}, \widehat{w}]$ and $\widehat{R}_{SIS}^{\pi}[\widehat{w}]$ in details, and we state the analysis as the following theorem.

**Theorem A.3.** *Let* $Var_{\mathcal{D}_{\pi_0}} \left[ \widehat{R}_{res}^{\pi}[\widehat{V}, \widehat{w}] \right]$ *be defined in Proposition 3.1, and suppose the normalization is constant (and hence an ordinary importance sampling which is unbiased). Then we can further break it into two terms*

$$Var_{\mathcal{D}_{\pi_0}} \left[ \widehat{R}_{res}^{\pi}[\widehat{V}, \widehat{w}] \right] = \frac{1}{n} \left( Var \left[ \widehat{w}(s)\varepsilon_{\widehat{V}}(s) \right] + \mathbb{E} \left[ \widehat{w}(s)^2 \left( \delta_1(s,a) + \gamma \delta_2(s,a,s') \right)^2 \right] \right), \qquad (18)$$

*where* $\varepsilon_{\widehat{V}}(s) = \widehat{V}(s) - r^{\pi}(s) - \gamma \mathcal{P}^{\pi}\widehat{V}(s')$ *is the Bellman residual(not the empirical one),* $\delta_1(s,a) = \frac{\pi(a|s)}{\pi_0(a|s)}r(s,a) - r^{\pi}(s)$ *is the randomness for action and* $\delta_2(s,a,s') = \frac{\pi(a|s)}{\pi_0(a|s)}\widehat{V}(s') - \mathcal{P}^{\pi}\widehat{V}(s)$ *is the randomness for transition operator over function* $\widehat{V}$. *Both* $\delta_1$ *and* $\delta_2$ *is zero mean if we condition over* $s$.

*Compared with* $Var \left[ \widehat{R}_{SIS}^{\pi}[\widehat{w}] \right]$ *we have:*

$$Var \left[ \widehat{R}_{SIS}^{\pi}[\widehat{w}] \right] = \frac{1}{n} (Var \left[ \widehat{w}(s)r^{\pi}(s) \right] + \mathbb{E} \left[ \widehat{w}(s)^2 \delta_1(s,a)^2 \right]) \qquad (19)$$

*Proof.* $\widehat{R}_{res}^{\pi}[\widehat{V}, \widehat{w}]$ can be written as

$$\widehat{R}_{res}^{\pi}[\widehat{V}, \widehat{w}] = \frac{1}{n} \sum \widehat{w}(s) \left( \beta_{\pi/\pi_0}(a|s)(r + \gamma\widehat{V}(s')) - \widehat{V}(s) \right),$$

where $\beta_{\pi/\pi_0}(a|s)$ is short for $\frac{\pi(a|s)}{\pi_0(a|s)}$. We can break $\beta_{\pi/\pi_0}(a|s)(r + \gamma\widehat{V}(s')) - \widehat{V}(s)$ into

$$\beta_{\pi/\pi_0}(a|s)(r(s,a) + \gamma\widehat{V}(s')) - \widehat{V}(s)$$
$$= (-\widehat{V}(s) + r^{\pi}(s) + \gamma\mathcal{P}^{\pi}\widehat{V}(s)) + \underbrace{(\beta_{\pi/\pi_0}(a|s)r(s,a) - r^{\pi}(s))}_{\delta_1} + \gamma\underbrace{(\beta_{\pi/\pi_0}(a|s)\widehat{V}(s') - \mathcal{P}^{\pi}\widehat{V}(s))}_{\delta_2}$$
$$= -\varepsilon_{\widehat{V}}(s) + \delta_1(s,a) + \gamma\delta_2(s,a,s').$$

where $\varepsilon_{\widehat{V}} = \widehat{V} - r^\pi - \mathcal{P}^\pi \widehat{V}$ is the Bellman residual and the if we condition over $s$ we have the expectations for $\delta_1$ and $\delta_2$ are 0. Also notice that if we condition over $s$ then $\varepsilon_{\widehat{V}}(s)$ become a constant thus it is independent to $\delta_1$ and $\delta_2$. Thus we have:

$$\text{Var}\left[\widehat{w}(s)\left(\beta_{\pi/\pi_0}(a|s)(r + \gamma\widehat{V}(s')) - \widehat{V}(s)\right)\right]$$
$$= \text{Var}\left[\widehat{w}(s)\left(-\varepsilon_{\widehat{V}}(s) + \delta_1(s,a) + \gamma\delta_2(s,a,s')\right)\right]$$
$$= \text{Var}\left[\widehat{w}(s)\varepsilon_{\widehat{V}}(s)\right] + \mathbb{E}\left[\widehat{w}(s)^2\left(\delta_1(s,a) + \gamma\delta_2(s,a,s')\right)^2\right]$$

Therefore we have:

$$\text{Var}\left[\widehat{R}_{\text{DR}}^\pi[\widehat{V}, \widehat{w}]\right] = \frac{(1-\gamma)^2}{n_0}\text{Var}[\widehat{V}(s_0)] + \frac{1}{n}\left(\text{Var}\left[\widehat{w}(s)\varepsilon_{\widehat{V}}(s)\right] + \mathbb{E}\left[\widehat{w}(s)^2\left(\delta_1(s,a) + \gamma\delta_2(s,a,s')\right)^2\right]\right).$$

For $\text{Var}\left[\widehat{R}_{\text{SIS}}^\pi[\widehat{w}]\right]$ we have:

$$\text{Var}\left[\widehat{R}_{\text{SIS}}^\pi[\widehat{w}]\right] = \frac{1}{n}\text{Var}\left[\widehat{w}(s)\beta_{\pi/\pi_0}(a|s)r(s,a)\right]$$
$$= \frac{1}{n}\text{Var}\left[\widehat{w}(s)r^\pi(s) + \widehat{w}(s)\delta_1(s,a)\right]$$
$$= \frac{1}{n}\left(\text{Var}\left[\widehat{w}(s)r^\pi(s)\right] + \mathbb{E}\left[\widehat{w}^2(s)\delta_1(s,a)^2\right]\right).$$

$\square$

From the theorem we can see that the variance of residual comes from two parts, the majority part relies on the variance of $|\varepsilon_{\widehat{V}}(s)|$ is usually much smaller than $r^\pi$ as the majority variance of state visitation importance sampling. When $\widehat{V} \approx V^\pi$, $\varepsilon_{\widehat{V}}(s) \approx 0$, the variance of residual only comes from $\delta_1(s,a)$ and $\delta_2(s,a,s')$.

In practice when we get trajectories data from $\pi_0$, to get samples uniformly from $d_{\pi_0}$, we can either draw sample $s_t$ depends on its discounted factor $\gamma^t$ or add an importance weight $\gamma^t$ as we did in our empirical estimator in equation (6) and equation (10).

## A.4 Proof of Theorem 4.1

**Theorem 4.1** (Primal Dual). *I) Define $w_{\rho/\pi_0}(s) = \frac{\rho(s)}{d_{\pi_0}(s)}$. We have*

$$L(V, \rho) = R_{DR}^\pi[V, w_{\rho/\pi_0}], \qquad \text{and hence} \qquad L(V, d_\pi) = L(V^\pi, \rho) = R^\pi, \ \forall\, V, \rho,$$

*which suggests that $L(V, \rho)$ is "doubly robust" in that it equals $R^\pi$ if either $V = V^\pi$ or $\rho = d_\pi$.*

*II) The primal problem* (14) *forms a strong duality with the following dual problem,*

$$R^\pi = \max_{\rho \geq 0}\left\{\sum_s \rho(s)r^\pi(s) \qquad s.t. \qquad \rho(s') = (1-\gamma)\mu_0(s') + \gamma\mathcal{T}^\pi\rho(s'), \quad \forall s'\right\}, \qquad (16)$$

*where $\mathcal{T}^\pi$ is defined in* (3).

*Proof.* The Lagrangian can be written as:

$$L(V, \rho) = (1-\gamma)\sum_s \mu_0(s)V(s) - \sum_s \rho(s)\left(V(s) - r^\pi(s) - \gamma\mathcal{P}^\pi V(s)\right)$$

$$= \underbrace{\sum_s (1-\gamma)\mu_0(s)V(s)}_{=R_{\text{VAL}}^\pi[V]} - \underbrace{\sum_s \rho(s)\left(I - \gamma\mathcal{P}^\pi\right)V(s)}_{=R_{\text{bridge}}^\pi[V, w_{\rho/\pi_0}]} + \underbrace{\sum_s \rho(s)r^\pi(s)}_{=R_{\text{SIS}}^\pi[w_{\rho/\pi_0}]}$$

$$= \sum_s (1-\gamma)\mu_0(s)V(s) - \sum_s \left(I - \gamma\mathcal{T}^\pi\right)\rho(s)V(s) + \sum_s \rho(s)r^\pi(s)$$

$$= \sum_s \left((1-\gamma)\mu_0(s) - (I - \gamma\mathcal{T}^\pi)\rho(s)\right)V(s) + \sum_s \rho(s)r^\pi(s).$$

We can see that the Lagrangian $L(V, \rho)$ is actually our doubly robust estimator $R_{\text{DR}}^{\pi}[V, w_{\rho/\pi_0}]$.

From the last equation we can derive our dual as:

$$\max_{\rho \geq 0} \quad \sum_s \rho(s) r^{\pi}(s)$$
$$\text{s.t.} \quad \rho(s) = (1 - \gamma)\mu_0(s) + \gamma \mathcal{T}^{\pi} \rho(s), \ \forall s.$$

$\square$

# B   DOUBLY ROBUST ESTIMATOR FOR AVERAGE CASE

## B.1   PRIMAL DUAL FRAMEWORK

We start from primal dual framework to get our doubly robust estimator similar to section 4. To estimate the average reward of a given policy $\pi$, we can consider solve the following linear programming:

$$\max_{\rho \geq 0} \sum_s \rho(s) r^{\pi}(s)$$
$$\text{s.t.} \quad \sum_s \rho(s) = 1, \quad \rho(s) = \mathcal{T}^{\pi} \rho(s), \ \forall s, \tag{20}$$

where $\rho(s)$ is the stationary distribution of states under $\mathcal{P}^{\pi}$, and the objective is the average reward given $\pi$.

Consider the Lagrangian of above linear programming:

$$L(V, \rho, \bar{v}) = \sum_s \rho(s) r^{\pi}(s) - \sum_s V(s)(\rho(s) - \mathcal{T}^{\pi}\rho(s)) - \bar{v}(\sum_s \rho(s) - 1)$$
$$= \bar{v} - \sum_s \rho(s)(V(s) - r^{\pi}(s) - \mathcal{P}^{\pi}V(s) + \bar{v}). \tag{21}$$

From Equation (21) we can get the dual formula as:

$$\min_{V, \bar{v}} \ \bar{v}$$
$$\text{s.t.} \quad \bar{v} + V(s) - \mathcal{P}^{\pi}V(s) - r^{\pi}(s) \geq 0, \ \forall s, \tag{22}$$

where $V(s)$ is the value function and $\bar{v}$ is the average reward we want to optimize.

Notice that in average case, $V^{\pi}(s)$ can be viewed as the fixed-point solution to the following Bellman equation:

$$V^{\pi}(s) - \mathbb{E}_{s', a|s \sim d_{\pi}}[V^{\pi}(s')] = \mathbb{E}_{a|s \sim \pi}[r(s, a) - \bar{v}].$$

Note that this explains the constraint and only if we pick $\bar{v} = R^{\pi}$, we can find a $V$ to guarantee the constraint $\bar{v} + V(s) - \mathcal{P}^{\pi}V(s) - r^{\pi}(s) \geq 0$ holds true.

## B.2   DOUBLY ROBUST ESTIMATOR

We want to build the doubly robust estimator via the lagrangian. However, the Lagrangian consist of three term $\rho, V$ and $\bar{v}$. It is counter-intuitive if we already given an estimator of $\bar{v} \approx R^{\pi}$ into our estimator.

A better way to solve this problem is to remove the constraint $\sum \rho(s) = 1$, but we divide it as an self-normalization. Then our Lagrangian becomes

$$L(V, \rho) = \frac{\sum_s \rho(s) r^{\pi}(s) - \sum_s V(s)(\rho(s) - \mathcal{T}^{\pi}\rho(s))}{\sum \rho(s)}.$$

which can be utilized to define the doubly robust estimator for average reward.

**Definition B.1.** *Given a learned value function $\widehat{V}(s)$ for policy $\pi$ and an estimated density ratio $\hat{w}(s)$ for $w_{\pi/\pi_0}(s)$, we define*

$$\widehat{R}_{DR}^\pi[\widehat{V}, \widehat{w}] := \frac{\sum_{s,a,r,s' \in \mathcal{D}} \widehat{w}(s) \left( \beta_{\pi/\pi_0}(a|s)(r + \widehat{V}(s')) - \widehat{V}(s) \right)}{\sum_{s \in \mathcal{D}} \widehat{w}(s)},$$

*where $\beta_{\pi/\pi_0}(a|s) = \frac{\pi(a|s)}{\pi_0(a|s)}$.*

Similarly to Theorem 3.1 we will have our double robustness for our average doubly robust estimator:

**Theorem B.2.** *Suppose we have infinite samples and we can get*

$$R_{DR}^\pi[\widehat{V}, \widehat{w}] = \frac{\mathbb{E}_{s \sim d_{\pi_0}} \left[ \widehat{w}(s) \left( r_\pi(s) - \widehat{V}(s) + \mathcal{P}^\pi \widehat{V}(s) \right) \right]}{\mathbb{E}_{s \sim d_{\pi_0}} [\widehat{w}(s)]}.$$

*Then we have*

$$R_{DR}^\pi[\widehat{V}, \widehat{w}] - R^\pi = \mathbb{E}_{s \sim d_{\pi_0}} \left[ \varepsilon_{\widehat{w}}(s) \varepsilon_{\widehat{V}}(s) \right], \tag{23}$$

*where $\varepsilon_{\widehat{V}}$ and $\varepsilon_{\widehat{w}}$ are errors of $\widehat{V}$ and $\widehat{w}$, respective, defined as follows*

$$\varepsilon_{\widehat{w}} = \frac{\widehat{w}(s)}{\mathbb{E}_{s \sim d_{\pi_0}}[\widehat{w}(s)]} - \frac{d_\pi(s)}{d_{\pi_0}(s)}, \qquad \varepsilon_{\widehat{V}}(s) = r_\pi - \widehat{V} + \mathcal{P}^\pi \widehat{V} - R^\pi.$$

*Proof.* A key observation is that

$$\begin{aligned}
\mathbb{E}_{s \sim d_{\pi_0}}[w_{\pi/\pi_0}(s)\varepsilon_{\widehat{V}}(s)] &= \mathbb{E}_{s \sim d_\pi}[r^\pi(s) - \widehat{V}(s) + \mathcal{P}^\pi \widehat{w}(s) - R^\pi] \\
&= (\mathbb{E}_{s \sim d_\pi}[r^\pi(s)] - R^\pi) + \mathbb{E}_{s \sim d_\pi}[-\widehat{V}(s) + \mathcal{P}^\pi \widehat{w}(s)] \\
&= 0
\end{aligned}$$

Thus we have

$$\begin{aligned}
R_{DR}^\pi[\widehat{V}, \widehat{w}] - R^\pi &= \mathbb{E}_{s \sim d_{\pi_0}} \left[ \frac{\widehat{w}(s)}{\mathbb{E}_{s \sim d_{\pi_0}}[\widehat{w}(s)]} \left( R^\pi + \varepsilon_{\widehat{V}}(s) \right) \right] - R^\pi \\
&= \mathbb{E}_{s \sim d_{\pi_0}} \left[ \frac{\widehat{w}(s)}{\mathbb{E}_{s \sim d_{\pi_0}}[\widehat{w}(s)]} \varepsilon_{\widehat{V}}(s) \right] \\
&= \mathbb{E}_{s \sim d_{\pi_0}} \left[ \frac{\widehat{w}(s)}{\mathbb{E}_{s \sim d_{\pi_0}}[\widehat{w}(s)]} \varepsilon_{\widehat{V}}(s) \right] - \mathbb{E}_{s \sim d_{\pi_0}}[w_{\pi/\pi_0}(s)\varepsilon_{\widehat{V}}(s)] \\
&= \mathbb{E}_{s \sim d_{\pi_0}} \left[ \varepsilon_{\widehat{w}}(s) \varepsilon_{\widehat{V}}(s) \right].
\end{aligned}$$

$\square$

Similar to discounted case we have $R_{DR}^\pi[\widehat{V}, \widehat{w}] = R^\pi$ if either $\widehat{w}$ or $\widehat{V}$ is accurate.

## C  EXPERIMENTAL DETAILS

### C.1  TABULAR CASE: TAXI

**Behavior and Target Policies Choosing**  We use an on-policy Q-learning to get a sequence of policy $\pi_0, \pi_1, ..., \pi_{19}$ as data size increases. We pick the last policy $\pi_{19}$ (almost optimum) as our target policy and $\pi_{18}$ as our behavior policy to guarantee that those policies are not far away. We set our discounted factor $\gamma = 0.99$.

**Train $\widehat{V}$ and $\widehat{\rho}$**  Separate from testing, we use a set of independent sample to first train a value function $\widehat{V}$ and a density function $\widehat{\rho}$. Both $\widehat{V}$ and $\widehat{\rho}$ have bias due to finite sample approximation.

For training $\widehat{V}$ and $\widehat{\rho}$, we choose to use Monte Carlo method to estimate $\widehat{V}$ and $\widehat{\rho}$. We first use the finite samples to get an estimated model $\widehat{T}(s'|s, a)$ and rewards function $\widehat{r}(s, a)$ and $\widehat{d}_0$. Then we solve the following linear equation (by iteration like power method, which is actually Monte Carlo):

$$\widehat{V}(s) = \sum_a \pi(a|s)\widehat{Q}^\pi(s, a),$$

$$\widehat{Q}^\pi(s, a) = \widehat{r}(s, a) + \gamma \sum_{s'} \widehat{T}(s'|s, a)\widehat{V}(s'),$$

$$\widehat{\nu}(s, a) = \widehat{\rho}(s)\pi(a|s),$$

$$\widehat{\rho}(s) = (1 - \gamma)\widehat{d}_0(s) + \gamma \sum_{s,a} \widehat{T}(s'|s, a)\widehat{\nu}(s, a).$$

**Estimate $R^\pi$ Using $\widehat{V}$ and $\widehat{\rho}$**  We put $\widehat{V}$ and $\widehat{\rho}$ into the Lagrangian as equation (15) as our doubly robust estimator. For those states we haven't visited, we set $\widehat{V}(s)$ and $\widehat{\rho}(s)$ as 0 and we self-normalized the $\widehat{\rho}$ to get a fair estimation.

### C.2  CONTINUOUS STATES OFF-POLICY EVALUATION

**Evaluation Environments**  We evaluate our method on two infinite horizon environments: Puck-Mountain and InvertedPendulum.

Puck-Mountain is an environment similar to Mountain-car, except that the goal of the task is to push the puck as high as possible in a local valley whose initial position is at the bottom of the valley. If the ball reaches the top sides of the valley, it will hit a roof and changes the speed to its opposite direction with half of its original speeds. The rewards was determined by the current velocity and height of the puck.

InvertedPendulum is a pendulum that has its center of mass above its pivot point. It is unstable and without additional help will fall over. We train a near optimal policy that can make the pendulum balance for a long horizon. For both behavior and target policies, we assume they are good enough to keep the pendulum balance and will never fall down until they reach the maximum timesteps. We use the implementation from OpenAI Gym (Brockman et al., 2016) and changing the dynamic by adding some additional zero mean Gaussian noise to the transition dynamic.

**Behavior and Target Policies Learning**  We use the open source implementation[2] of deep Q-learning to train a $32 \times 32$ MLP parameterized Q-function to converge. We then use the softmax policy of learned the Q-function as the target policy $\pi$, which has a default temperature $\tau = 1$. For the behavior policy $\pi_0$, we set a relative large temperature which encourages exploration. We set the temperature of the behavior policy $\tau_0 = 1.88$ for Puck-Mountain and $\tau_0 = 1.50$ for InvertedPendulum respectively.

**Training of density ratio $\hat{w}(s)$ and value function $\hat{V}(s)$**  We use a seperate training dataset with 200 trajectories whose horizon length is 1000 to learn the density ratio $\hat{w}(s)$ and the value function

---

[2]https://github.com/openai/baselines

---

**Algorithm 2** Optimization of density ratio $\widehat{w}$

---

**Input**: Transition data $\mathcal{D} = \{s_i, a_i, s'_i, r_i\}_{i=1}^n$ from the behavior policy $\pi_0$; a target policy $\pi$ for which we want to estimate the expected reward. Discount factor $\gamma \in (0,1)$, starting state $\mathcal{D}_0 = \{s_j^{(0)}\}_{j=1}^m$ from initial distribution.

**Initial** the density ratio $w(s) = w_\theta(s)$ to be a neural network parameterized by $\theta$, $f(s) = f_\beta(s)$ to be a neural network parameterized by $\beta$. We need to ensure that the final layer of $\theta$ is a softmax layer.

**for** iteration = 1,2,...,T **do**

    Randomly choose a batch $\mathcal{M} \subseteq \{1, \ldots, n\}$ uniformly from the transition data $\mathcal{D}$ and a batch $\mathcal{M}_0 \subseteq \{1, \ldots, m\}$ uniformly from start states $\mathcal{D}_0$.

    **for** iteration = 1,2,..., K **do**

        **Update** the parameter $\beta$ by $\beta \leftarrow \beta + \epsilon_\beta \nabla_\beta \hat{L}(w_\theta, \phi_\beta)$, where

$$\hat{L}(w_\theta, f_\beta) = \frac{1}{|\mathcal{M}|} \sum_{i \in \mathcal{M}} \left( (w(s'_i) - \gamma w(s_i) \frac{\pi(a_i|s_i)}{\pi_0(a_i|s_i)} - \frac{1}{2} f(s'_i)) f(s'_i) \right) - (1-\gamma) \frac{1}{|\mathcal{M}_0|} \sum_{j \in \mathcal{M}_0} f(s_j)$$

    **end for**

    **Update** the parameter $\theta$ by $\theta \leftarrow \theta - \epsilon_\theta \nabla_\theta \hat{L}(w_\theta, f_\beta)$.

**end for**

**Output**: the density ratio $\widehat{w} = w_\theta$.

---

**Algorithm 3** Optimization of value function $\widehat{V}$

---

**Input**: Transition data $\mathcal{D} = \{s_i, a_i, s'_i, r_i\}_{i=1}^n$ from the behavior policy $\pi_0$; a target policy $\pi$ for which we want to estimate the expected reward. Discount factor $\gamma \in (0,1)$.

**Initial** the value function $V(s) = V_\phi(s)$ to be a neural network parameterized by $\phi$, $f(s) = f_\beta(s)$ to be a neural network parameterized by $\beta$.

**for** iteration = 1,2,...,T **do**

    Randomly choose a batch $\mathcal{M} \subseteq \{1, \ldots, n\}$ uniformly from the transition data $\mathcal{D}$.

    **for** iteration = 1,2,..., K **do**

        **Update** the parameter $\beta$ by $\beta \leftarrow \beta + \epsilon_\beta \nabla_\beta \hat{L}(V_\phi, \phi_\beta)$, where

$$\hat{L}(V_\phi, f_\beta) = \frac{1}{|\mathcal{M}|} \sum_{i \in \mathcal{M}} \left( \left( V_\phi(s_i) - \frac{\pi(a_i|s_i)}{\pi_0(a_i|s_i)} (r_i + \gamma V_\phi(s'_i)) \right) f_\beta(s_i) - \frac{1}{2} f_\beta(s_i)^2 \right)$$

    **end for**

    **Update** the parameter $\phi$ by $\phi \leftarrow \phi - \epsilon_\phi \nabla_\phi \hat{L}(V_\phi, f_\beta)$.

**end for**

**Output**: the density ratio $\widehat{V} = V_\phi$.

---

$\hat{V}(s)$. For the training of density ratio, we adapt the algorithm 2 in Liu et al. (2018a) to train a neural network parameterized $w_\theta(s)$. Instead of taking the test function $f(s)$ into an RKHS $\mathcal{H}_\mathcal{K}$, we parameterize the test function $f(s) = f_\beta(s)$ to be a neural network with parameter $\beta$, and perform minimax optimization over parametr $\theta$ and $\beta$. A detail description can be found in Algorithm 2.

Similarly, for the training of value function, we use primal-dual based optimization methods (Dai et al., 2018; Feng et al., 2019) to minimize the bellman residual:

$$\min_\phi \max_{f_\beta \in \mathcal{F}} \frac{1}{|\mathcal{M}|} \sum_{i \in \mathcal{M}} \left( \left( V_\phi(s_i) - \frac{\pi(a_i|s_i)}{\pi_0(a_i|s_i)} (r_i + \gamma V_\phi(s'_i)) \right) f_\beta(s_i) - \frac{1}{2} f_\beta(s_i)^2 \right),$$

where $V_\phi(s)$ is the parameterized value function and $f_\beta(s)$ is the test function. We also perform minimax over parameter $\phi$ and $\beta$. A detail description can be found in Algorithm 3.

For the network structures, we use $32\times32$ feed forward neural networks to parameterize value function $V_\phi$ and density ratio $w_\theta(s)$, and we use one hidden neural network with 10 units to parameterize the test function $f_\beta(s)$. We use Adam Optimizer for all our experiments.

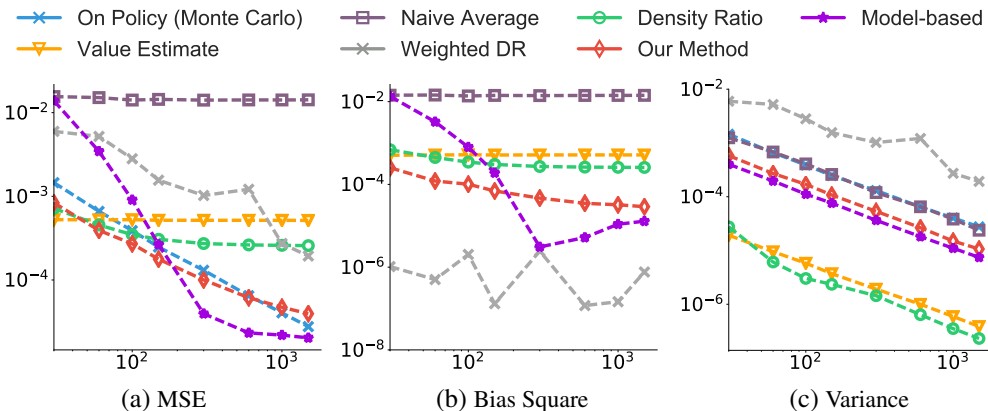

Figure 4: Additional results on Taxi, with an extra curve for model based method.

**Estimate $R^\pi$ using $\widehat{V}$ and $\widehat{w}$**    Given data samples from the policy $\pi_0$, We can directly use $\widehat{R}^\pi_{\text{DR}}$ in equation (11) to estimate $R^\pi$.

## D   ADDITIONAL EXPERIMENTAL RESULTS

We add a standard model-based baseline in Figure 4 following the same procedure as Liu et al. (2018a).[3] Figure 4 shows that the performance of the model-based approach largely depends of the size of the data, which is consistent with that in Liu et al. (2018a).

---

[3]https://github.com/zt95/infinite-horizon-off-policy-estimation.

