# OpenReview forum: "Doubly Robust Bias Reduction in Infinite Horizon Off-Policy Estimation"
_ICLR.cc/2020/Conference — Accept (Spotlight)_

### Official Review · AnonReviewer2 · 2019-10-22
**Official Blind Review #2**

**Rating:** 8

**Review:**

  *Synopsis*:
  This paper provides a new doubly robust estimator for off-policy policy evaluation, based on the new infinite horizon technique (i.e. using an estimate of the state density ratio as opposed to long products of action importance weights). They show the doubly robust estimator's bias is dependent on a product of the error of the value function estimate and stationary distribution ratio estimate, which provides improvements over the initial infinite horizon estimator. They also provide some nice discussion of the relation of their method and Lagrangian duality, which was quite interesting and insightful. Finally, the paper shows the usual empirical comparisons.

  Main Contributions:
  - Doubly robust estimator for infinite horizon off-policy policy evaluation

  *Review*:
  Overall, I quite like this paper and think the quality is at a high level. The proposed doubly robust estimator is well supported theoretically and empirically and improves on the prior art. I am recommending a weak accept for this paper as it will be a nice contribution for the community, but I have some clarifications/updates I would like to see to improve the readability of the paper (specifically C4).

  I also have a few questions, and clarifications that I would like the authors to address during the rebuttal period.

  *Clarifications*:
  C1: In the section "Off-policy state visitation importance sampling", doesn't equation 4 involve an expectation over the target policy? Or am I missing something here?

  C2: The bias of the estimator decreases as our estimate of the density ratio and value function improves. It might be useful to more clearly compare this bias to the original estimator proposed by Liu.

  C3: Proof of theorem 3.1: I would like you to clearly finish the proof. I think clarity and completeness in appendices is importance over conciseness as there is usually no limit on pages.

  C4: The proof of theorem 3.2 is not obviously linked to theorem A.3. The proof for theorem 3.2 should be clearly stated and included in the appendix, without an unstated implicit link to A.3. This will make your theoretical analysis more clearly understandable, and expendable for future work. I will happily increase my score if this is clarified in a future version. I also will decrease my score if this is not clarified.

  *Competitors*

  - You may want to include a model based approach, just for completeness (as in the infinite horizon paper from Liu).

  *Questions*

   Q1: I'm curious about the difference between using an ANN for estimating the density ratio (as opposed to a kernel method). Have you run experiments with the kernel method proposed in Liu's paper? While I don't think it is necessarily needed for the paper to be complete, I think the difference would be interesting to see what factors contribute to the algorithm's better performance.


  Minor typos not taken into account for the review:
  - Section 3.1 "it is useful to exam the": exam -> examine
  - Section 4: "hence yielding an true expected reward": an -> a
  - Section 4: "it is natural to intuitize": intuitize -> intuit?
  - The sentence right before section 4 could use a rewrite.
  - For readability it would be useful to include theorem statements ahead of proofs in the appendix.
  - Theorem A.3: do you mean with \hat{R} defined in 3.2 (not the variance as this is not defined in 3.2).

Edit:

- Due to the author rebuttal and updates to the paper I've increased my score.

**Experience Assessment:**

I have read many papers in this area.

**Review Assessment: Checking Correctness Of Derivations And Theory:**

I carefully checked the derivations and theory.

**Review Assessment: Checking Correctness Of Experiments:**

I carefully checked the experiments.

**Review Assessment: Thoroughness In Paper Reading:**

I read the paper thoroughly.

---

> ### Author Response · Authors · 2019-11-12
> **Response to Review #2**
>
> We thank Reviewer #2 for the valuable comments and suggestions. We have updated a version based on your feedback, where the more important changes are highlighted in red.
> Below are responses to your specific questions:
>
> - 1. Equation 4 requires an on-policy expectation
>
> Equation 4 is in fact in the subsection “Estimation via State Density Function”.  It gives an alternative yet equivalent expression for the policy value, and is the basis for the next subsection “Off-Policy State Visitation Importance Sampling” (e.g., compare to equation 5).
>
>
> -2. Comparison of the Bias of the work proposed by Liu. et al.
>
> We did compare the bias of the work proposed by Liu. et al. See the discussion for the paragraph after Theorem 3.1, where we list the result for $R^\pi_{\text{SIS}}[\hat{w}] - R^\pi$ can be written as the error of $\mathbb{E}_{d_{\pi_0}}[ \hat{w}(s) r^\pi(s) ]$. We have updated the proof for Theorem 3.1 in appendix which includes the bias analysis also for $R^\pi_{\text{SIS}}$ and $R^\pi_{\text{VAL}}$.
>
>
> -3. Completeness of Theorem 3.1.
>
> The revised version now expanded the original proof (still in Appendix A.2).  The new proof includes more details and should make the proof clearer.  We also repeated the theorem statement in the appendix for the reader’s convenience.
>
>
> -4. Clarification of theorem 3.2 (now Proposition 3.1).
>
> Thank you for the great suggestion.  The new version gives a complete proof for Proposition 3.1 (Appendix A.3), and also connects it to Theorem A.3 explicitly.
>
> i) The proof of Theorem 3.2 is simply to break the variance into a sum of variances of two independent terms, $\hat{R}_{\text{VAL}}^\pi$ and $\hat{R}_{\text{res}}^\pi$.  Since the proof is simple, we change the theorem into a proposition.
> ii) Link for Theorems 3.2 and A.3: The proof of Theorem 3.2 decomposes the variance of DR into two terms (see (i) above), while Theorem A.3 focuses on the second term, $\rm{Var}[\hat{R}_{\text{res}}^\pi[\hat{V}, \hat{w}]]$ by further analyzing the different sources of randomness.
>
>
> - 5. Comparing with model based approach
>
> We added the additional results of the taxi environment with a model-based baseline in Figure 4 (see appendix D). General model-based estimators suffer from data with a small sample size where they don’t have enough samples for each transition. The observation is similar to the result in Liu et al.
>
>
> -6. ANN vs kernel methods in Optimization.
>
> We would like to clarify that for these two methods, we use a neural network to parameterize the density ratio function. The difference between these two methods is the choice of function class for the test function $f$: the method introduced in the appendix uses neural networks, while Liu et. al choose a RKHS.  Both methods can be viewed as alternatives for solving the minimax saddle-point problem (equation (15) in Liu et. al), with different choices of function classes for the dual variables in the minimax problem.
>
> These two methods can be both efficient to learn the density ratio if we carefully design the function class and tune the hyperparameters. For the kernel method mentioned in Liu et.al, we can easily get rid of the inner maximization but it requires us to choose a proper ISPD kernel (e.g., Gaussian kernel with a proper bandwidth). For the method we mentioned in the appendix, it requires less hyperparameter tuning while it needs to perform additional inner maximization on an additional auxiliary neural network, which can be unstable in some scenarios. However, for empirical experiments in our paper, we did not observe a significant performance gap between these two methods when we carefully tune the hyperparameters.
>
> Finally, we would like to emphasize that it is not our main contribution to propose new algorithms for optimizing density ratio or value functions. Instead, our focus is on the new DR estimator and its double robustness property. We will include a thorough discussion on related optimization techniques in a future version.
>
>
> -7. Typo
> Thanks for pointing out the typos, which have been fixed in the revision.

---

> > ### Comment · AnonReviewer2 · 2019-11-13
> > **Thank you for the response**
> >
> > Thanks for the response, and the updates to the paper. You have clarified all the questions and concerns I mentioned well, and I think the paper is much clearer. I also appreciate the addition of the model based comparison as a benchmark.
> >
> > I'm quite happy with the updates and will increase my score.

---

> > > ### Author Response · Authors · 2019-11-14
> > > **Thank you**
> > >
> > > Thank you for raising the score! We will make the paper more clear in the final version.

---

### Official Review · AnonReviewer1 · 2019-10-23
**Official Blind Review #1**

**Rating:** 8

**Review:**

Comments :

This paper provides an approach for reducing bias in long horizon off-policy evaluation (OPE) problems, extending recent work from Liu et al., 2018 that estimates the ratio of the stationary state distributions in off-policy evaluation for reducing variance. The paper provides a doubly robust method for reducing bias, since it requires separately estimating a value function. The key idea of the paper is to provide low variance, low bias OPE since their approach relies on accurately estimating the state distribution ratio and also the estimation of the value function.

The paper provides a follow up on Liu et al., 2018 and other recent works in off-policy learning that tries to estimate the state distribution ratio directly, but can introduce high inaccuracy if the obtained ratio estimates are inaccurate. In line of that, this approach seems useful as it tries to reduce the error from inaccurate ratio estimates by incorporating prior works with an additional value function estimation.

Furthermore, the paper introduces a new perspective to doubly robust estimation that tries to reduce bias, instead of variance in previously known OPE literature (Jiang et al., 2016; Thomas et al., 2016). It is interesting to see how doubly robust can be related to primal-dual relations and the connections between these approaches, which is a novel contribution to the best of my knowledge.

- The fundamental connection comes from observing OPE withdoubly robust estimator that estimates \hat{V} and equation 6 that incorporates the ratio of the state density. This is clearly written in equations 7 and 8, that are two ways of evaluating the value function with off-policy samples known in the literature.

- It uses the property of the Bellman recursive expression for the estimated value function of doubly robust OPE estimator and uses it in the OPE with state density ratio, leading to equation 9, and obtaining the bridge estimator that now relies on both estimates of value function \hat{V} and the state density ratio. Although initially this does not seem useful, but the authors clarify this and how a bias reducting method can be achieved as in equation 11.

- It is a nice and elegant trick, exploiting the connections between OPE estimators leading to a bias reduction method that seems quite interesting.

- The most interesting part of the paper comes from section 4 that exploits the connection between doubly robust approaches with Lagrangian duality, and that their approach is equivalent to a primal dual formulation of policy evaluation. This stands out in itself as a novel contribution of the paper. Equations 14 and 15 best writes down the connections, as to how policy evaluation can be formualted as a Lagrangian function for optimization.

- Although the authors point out how equation 15 is equivalent to equation 11 - this does not seem straight forward at first unless carefully looked at. I would encourate the authors to perhaps add more clarity that exploits this connection, to make the paper more self-contained.

- In terms of experiments, the paper compares the doubly robust approach with previous works that estimates the density ratio, along with other baseline comparisons such as weighted DR. In both the two evaluated problems, their approach seems useful in terms of obtaining better accuracy (lower MSE).

- However, I am not sure to what extent this approach can be scaled to more complicated tasks for OPE? Are there any example domains where the proposed method fails, or is difficult to scale up to more complicated tasks?

- The current set of experiment results seems adequate, given the theoretical contribution and that most OPE papers evaluate on such domains. But overall, it might be useful to analyse the significance of this approach when scaling to more complicated domains.


Overall, I think the paper has a neat and elegant theoretical contribution, exploiting the connection of OPE with primal-dual frameworks that seems quite novel to me. Experimental set of results are properly presented too showing significance of the approach compared to previously known baselines. I think such papers exploting connections with other literature is useful for the community in general, and this paper has significantly novel theoretical contribution. Hence, I would recommend for acceptance of this paper.



**Experience Assessment:**

I have published one or two papers in this area.

**Review Assessment: Checking Correctness Of Derivations And Theory:**

I assessed the sensibility of the derivations and theory.

**Review Assessment: Checking Correctness Of Experiments:**

I assessed the sensibility of the experiments.

**Review Assessment: Thoroughness In Paper Reading:**

I read the paper thoroughly.

---

> ### Author Response · Authors · 2019-11-12
> **Response to Review #1**
>
> We thank Reviewer #1 for the valuable comments and suggestions. We have updated a version based on your feedback, where the more important changes are highlighted in red.
> Below are responses to your specific questions:
>
> - 1. Equivalent between equation (15) and equation (11) .
>
> The main difference of equation (15) and equation (11) is that equation (15) uses a density function $\rho(s)$ which is equal to $d_{\pi_0}(s)w(s)$ in previous section for the density ratio function $w(s)$. We rewrote equation (11) in an explicit form that makes the connection clearer.
>
>
> -  2.  Generalization of DR to more complicated tasks for OPE.
>
> Although this paper focuses on theoretical and algorithmic contributions, we have evaluated the proposed estimator on benchmarks that are popularly used in the recent literature of off-policy evaluation.  That said, we appreciate the suggestion, and are indeed interested in a more systematic empirical evaluation across more tasks with varying complexity.

---

> > ### Comment · AnonReviewer1 · 2019-11-13
> > **Thank you; Open-Source Implementation would make the paper extremely useful!**
> >
> > Thank you for your response.
> >
> > As for (2) on generalization of DR to more complicated tasks - I do agree, and understand that this may perhaps be beyond the scope of the paper. It would be useful to add more empirical analysis, but certainly not needed, given evaluation is being already made on more of the standard OPE tasks.
> >
> > One more comment : There are several recent progress in OPE and correcting for the state distributions in OPE. It would be very useful if you release the code of your paper, and the benchmarks used - as the recent progress in OPE methods will certainly benefit the community.
> >
> > There are several benchmark evaluations available for many RL algorithms, but the open-source implementations on OPE benchmarks is certainly lacking to my knowledge. Open-sourcing the code base for the experiments would help the community a lot! :)

---

> > > ### Author Response · Authors · 2019-11-14
> > > **Thank you for the suggestion!**
> > >
> > > Thank you for the great suggestion. We will release the code and related benchmarks shortly after the acceptance of the paper.

---

### Official Review · AnonReviewer3 · 2019-10-23
**Official Blind Review #3**

**Rating:** 6

**Review:**

This paper proposes a new algorithm for the off-policy evaluation problem in reinforcement learning. It combines the value function learning method and the stationary distribution ratio estimators. The proposed method achieves double robustness, which means the proposed estimator is consistent as long as the value function or ratio estimator is consistent. Empirical results on some control domains are presented to verify the effectiveness of the algorithm.

I think this paper has some nice contribution to the area, by introducing a doubly robust estimator based on the density ratio, and also a new idea to achieve double robustness. I will vote for accept, but I think there is room for improvement of this paper.

Detailed comments:
 - The proposed estimator is not using control variate but using dual structure between value function and stationary distribution ratio, which is a novel idea comparing with similar doubly robust estimators.
 - Theorem 3.2, or at least the way it is presented, is less intuitive and makes me confused. If the variance of the DR estimator is always larger than the variance of value function, why I should use this estimator instead of value function. If the argument is the MSE of value function is potentially larger due to the bias. Then an effective analysis would be about MSE instead of variance.
 - If I have an oracle of density ratio, is the doubly robust estimator still unbiased, which is generally true for DR using control variate? This would be an important point to compare this work with control variate methods.
 - Very recent work https://arxiv.org/abs/1908.08526 also proposes a doubly robust estimator in similar settings. It's worth to mention it in the related work.

**Experience Assessment:**

I have published one or two papers in this area.

**Review Assessment: Checking Correctness Of Derivations And Theory:**

I carefully checked the derivations and theory.

**Review Assessment: Checking Correctness Of Experiments:**

I assessed the sensibility of the experiments.

**Review Assessment: Thoroughness In Paper Reading:**

I read the paper thoroughly.

---

> ### Author Response · Authors · 2019-11-12
> **Response to Review #3**
>
> We thank Reviewer #3 for the valuable comments and suggestions. We have updated a version based on your feedback, where the more important changes are highlighted in red.
> Below are responses to your specific questions:
>
> - 1. Clarification on Theorem 3.2 (Proposition 3.1 in the revised version).
>
> As we stated in the paper, the variance of the doubly robust estimator is not guaranteed to be lower than decrease compared to previous estimators. The main difficulty of the analysis is that we have two different sources of samples, $\mu_0$ and $d_{\pi_0}$, which may not be independent in general.  Theorem 3.2 is for the simplified situation where $\mu_0$ and $d_{\pi_0}$ are assumed to be independent.  In practice, however, the actual variance of DR can be lower than what the theorem states, as discussed in the paragraph following Proposition 3.1.
>
>
> - 2. If the DR estimator has an oracle of density ratio, is the DR unbiased?
>
> Yes.  This is a direct consequence of Thm 3.1: when the correct density ratio is used, $\varepsilon_{\hat{w}} = 0$, implying the bias of DR (i.e., RHS of (12)) is 0.  We discussed it in the paragraph right after Thm 3.1, and have made it more explicit.
>
>
> - 3. Related work on Doubly Robust off-policy evaluation on infinite RL
>
> Thanks for suggesting the related and interesting paper, https://arxiv.org/abs/1908.08526 .  The authors considered *marginalized* importance sampling, where an importance ratio function is estimated for each step in an episode.  In contrast, we consider importance sampling in the *stationary/invariant* distribution, where a single importance ratio function is shared by all steps in an infinite-length trajectory.  We have updated the related work section accordingly.

---

> > ### Comment · AnonReviewer3 · 2019-11-13
> > **Thanks the author for the response**
> >
> > Thanks the author for the response.
> >
> > 1. One more follow-up question about theorem 3.2. According to your response, it seems that theorem 3.2 assume \pi and \mu are *independent*, because it's easy to analysis, but make the variance looks large. In practice, due to \pi and \mu are *not independent*, the variance can be lower than the theorem. It's interesting to learn this. In the dependent case, does the variance of DR smaller than the variance of value function? Could you provide some intuition on why the dependent case seems to be a better case for DR? I understand it's could be technically hard to provide formal proof.
> >
> > 2. For the related work, sorry it's my mistake. I think the same authors (Kallus & Uehara) have another work with a similar title (https://arxiv.org/abs/1909.05850), which is using the stationary ratio. I actually meant this one.

---

> > > ### Author Response · Authors · 2019-11-14
> > > **Response to the Follow-up Questions**
> > >
> > > We thank Reviewer #3 for the follow-up comments. Below are responses to your specific questions:
> > >
> > > - 1.Variance analysis
> > >
> > > In general, if $R_{\mathrm{res}}$ and $R_{\mathrm{VAL}}$ are negatively correlated for the joint distribution of $\mu_0$ and $d_{\pi_0}$, we can reduce the variance. We leave it as future work to study the precise conditions under which it holds.
> > >
> > > On the other hand, we do not claim DR has a lower variance than VAL, which does not seem to be true in general.  This can be seen in the special case of contextual bandits.  Our discussion following Proposition 3.1 is mainly to compare DR and SIS.
> > >
> > >
> > > - 2.Related work
> > >
> > > Thanks for pointing out this interesting paper which gives a variant of our estimator (they replace V value function with Q action-value function). The paper is mostly theoretical, and it's interesting to have an experimental comparison in future work. We have cited the paper in the related work, see our newest version.

---

### Author Response · Authors · 2019-11-12
**General Reponse to All Reviewers**

We thank all the reviewers for the valuable comments and suggestions. We have updated a version of our paper in the following revisions (with important changes highlighted in red):

1. We rewrote the variance analysis part based on the suggestions by reviewers #2 and #3.
2. We added additional related works, including the one mentioned by reviewer #2.
3. We improved the proofs in the appendix by restating the theorems to make it self-contained. And we added more discussion for bias and variance part.
4. We changed equation (11) into an explicit form based on a suggestion of reviewer #1.
5. We added a model-based comparison experiment in appendix D.
6. We fixed the typos reviewers mentioned.

---

### Decision · Program_Chairs · 2019-12-19

**Decision:**

Accept (Spotlight)

**Comment:**

The paper proposes a doubly robust off-policy evaluation method that uses both stationary density ratio as well as a learned value function in order to reduce bias.
The reviewers unanimously recommend acceptance of this paper.